

# The complete mitogenome of *Arion vulgaris* Moquin-Tandon, 1855 (Gastropoda: Stylommatophora): mitochondrial genome architecture, evolution and phylogenetic considerations within Stylommatophora

Özgül Doğan[1,2], Michael Schrödl[2,3,4] and Zeyuan Chen[2,3]

[1] Department of Molecular Biology and Genetics, Faculty of Science, Sivas Cumhuriyet University, Sivas, Turkey
[2] SNSB-Bavarian State Collection of Zoology, Munich, Germany
[3] Department Biology II, Ludwig-Maximilians-Universität, Munich, Germany
[4] GeoBio-Center LMU, Munich, Germany

Corresponding author
Özgül Doğan,
odogan@cumhuriyet.edu.tr

## ABSTRACT

Stylommatophora is one of the most speciose orders of Gastropoda, including terrestrial snails and slugs, some of which are economically important as human food, agricultural pests, vectors of parasites or due to invasiveness. Despite their great diversity and relevance, the internal phylogeny of Stylommatophora has been debated. To date, only 34 stylommatophoran mitogenomes were sequenced. Here, the complete mitogenome of an invasive pest slug, *Arion vulgaris* Moquin-Tandon, 1855 (Stylommatophora: Arionidae), was sequenced using next generation sequencing, analysed and compared with other stylommatophorans. The mitogenome of *A. vulgaris* measures 14,547 bp and contains 13 protein-coding, two rRNA, 22 tRNA genes, and one control region, with an A + T content of 70.20%. All protein coding genes (PCGs) are initiated with ATN codons except for *COX1, ND5* and *ATP8* and all are ended with TAR or T-stop codons. All tRNAs were folded into a clover-leaf secondary structure except for *trnC* and *trnS1* (AGN). Phylogenetic analyses confirmed the position of *A. vulgaris* within the superfamily Arionoidea, recovered a sister group relationship between Arionoidea and Orthalicoidea, and supported monophyly of all currently recognized superfamilies within Stylommatophora except for the superfamily Helicoidea. Initial diversification time of the Stylommatophora was estimated as 138.55 million years ago corresponding to Early Cretaceous. The divergence time of *A. vulgaris* and *Arion rufus* (Linnaeus, 1758) was estimated as 15.24 million years ago corresponding to one of Earth's most recent, global warming events, the Mid-Miocene Climatic Optimum. Furthermore, selection analyses were performed to investigate the role of different selective forces shaping stylommatophoran mitogenomes. Although purifying selection is the predominant selective force shaping stylommatophoran mitogenomes, six genes (*ATP8, COX1, COX3, ND3, ND4* and *ND6*) detected by the branch-specific aBSREL approach and three genes (*ATP8, CYTB* and *ND4L*) detected by codon-based BEB, FUBAR and MEME approaches were exposed to diversifying selection. The positively selected substitutions at the mitochondrial PCGs of stylommatophoran

species seems to be adaptive to environmental conditions and affecting mitochondrial ATP production or protection from reactive oxygen species effects. Comparative analysis of stylommatophoran mitogenome rearrangements using MLGO revealed conservatism in Stylommatophora; exceptions refer to potential apomorphies for several clades including rearranged orders of *trnW-trnY* and of *trnE-trnQ-rrnS-trnM-trnL2-ATP8-trnN-ATP6-trnR* clusters for the genus *Arion*. Generally, tRNA genes tend to be rearranged and tandem duplication random loss, transitions and inversions are the most basic mechanisms shaping stylommatophoran mitogenomes.

## INTRODUCTION

Gastropoda is the most speciose class of Mollusca, including snails and slugs with very diverse feeding habits and a wide range of habitats (*Barker, 2009*). The about 63,000 gastropod species represent 476 families (*Bouchet et al., 2017*) and radiated in marine, freshwater and terrestrial ecosystems with detritivorous, herbivorous, carnivorous, predatory or parasitic life styles (*Ponder & Lindberg, 1997*). Most of the terrestrial gastropods are stylommatophoran pulmonates, with approximately 30,000 species distributed from polar to tropical regions (*Mordan & Wade, 2008*). Stylommatophorans are economically important as human food and because of their status of being major agricultural pests and/or vectors of parasites and invasiveness (*Barker, 2009*). The origin of Stylommatophora is within panpulmonate heterobranchs (*Jörger et al., 2010*) and the monophyly of the order is undisputed. Internal phylogenetic relationships of stylommatophorans were poorly resolved based on morphology but then investigated molecularly in different sampling sets of taxa with various methods and basically relatively short sequences. *Tillier, Masselmot & Tillirt (1996)* used the D2 region of *28S rRNA* to explore the phylogenetic relationships of pulmonates including a few stylommatophoran species; however, they reported that these short sequences would not have sufficient resolving power for investigating the relationships owing to the probable rapid radiation of pulmonate species. *Wade, Mordan & Clarke (2001)* and *Wade, Mordan & Naggs (2006)* presented more comprehensive molecular phylogenies based on relatively longer sequence information of the rRNA gene-cluster using 104 species (*Wade, Mordan & Clarke, 2001*) and 160 species (*Wade, Mordan & Naggs, 2006*) from Stylommatophora. Although these phylogenetic reconstructions accurately supported the monophyly of achatinoid and non-achatinoid clades, some clades of families that traditionally have been assumed to be monophyletic and some of the morphological groups based on excretory system; in particular, monophyly of some families and morphological groups were not supported.

The emergence and divergence time of Stylommatophora is also doubtful due to the fragmentary fossil records. The earliest land snails identified as stylommatophoran species are from upper Carboniferous and Permian but their classification has still

been controversial (*Solem & Yochelson, 1979*; *Hausdorf, 2000*). *Bandel (1991)* and *Roth et al. (1996)* suggested the oldest known fossil records from late Jurassic and Early Cretaceous (*Cheruscicola*) and Early Cretaceous (Pupilloidea). *Tillier, Masselmot & Tillirt (1996)* inferred that the Stylommatophora emerged in the transition between late Cretaceous and Paleocene (65–55 Ma) congruent with fossil records, based on the molecular data. However, all of the previous molecular dating analyses on Stylommatophora have been performed either with limited numbers of taxa or molecular markers (*Tillier, Masselmot & Tillirt, 1996*; *Jörger et al., 2010*; *Dinapoli & Klussmann-Kolb, 2010*; *Zapata et al., 2014*), therefore there is a need for further investigations in a more comprehensive sampling using more markers for better understanding of the phylogeny and timing of evolution of Stylommatophora.

In recent years, there has been a rapid increase in the number of sequenced mitochondrial genomes (mitogenomes) in parallel to revolution on high throughput DNA sequencing technology and data mining, providing a powerful tool for phylogenetic analysis (*Moritz, Dowling & Brown, 1987*; *Boore, 1999*; *Bernt et al., 2013a*). Animal mitogenomes are double-stranded circular molecules which are ∼16 kb in length and contain 13 protein coding genes (PCGs) forming the respiratory chain complexes: Complex I or NADH: ubiquinone oxidoreductase contains seven subunits of NADH dehydrogenase (*ND1–6* and *ND4L*), complex III or ubiquinol: cytochrome *c* oxidoreductase consists of cytochrome b (*CYTB*), complex IV or cytochrome *c* oxidase comprises three subunits of cytochrome c oxidase (*COX1–COX3*) and complex V or ATP synthase includes two subunits of the ATPase (*ATP6* and *ATP8*). The mitochondrial PCGs have generally been supposed to be evolving under neutral or nearly neutral selection (*Ballard & Kreitman, 1995*). Although it has been suggested that these genes are likely to be under strong purifying selection considering their functional importance, the selective pressures might vary even among closely related species and be influenced by environmental conditions (*Meiklejohn, Montooth & Rand, 2007*). The mitogenomes also encode the small and large subunit rRNAs (*rrnL* and *rrnS*) and twenty-two tRNA genes for the translation process of PCGs. In general, they harbour a single large non-coding region containing control elements necessary for replication and transcription (*Boore, 1999*). Mitogenomes have become widely used tools in recent phylogeny, phylogeography and molecular dating analyses in various taxa, because of their (1) relatively small size, (2) the high copy number, (3) maternal inheritance type and (4) relatively rapid rate of evolutionary change (*Moritz, Dowling & Brown, 1987*; *Gray, 1989*). The sequence information of mitogenomes has also been used in reconstructing phylogenies of several taxonomic groups within/including Gastropoda (*White et al., 2011*; *Stöger & Schrödl, 2013*; *Sevigny et al., 2015*; *Uribe et al., 2016a*; *Uribe et al., 2016b*; *Romero, Weigand & Pfenninger, 2016*; *Yang et al., 2019*). Although there have been some criticisms about the usage of mitogenomes in construction of gastropod phylogeny because of long branch attraction, substitution saturation and strand-specific skew bias (*Stöger & Schrödl, 2013*), within the recently diversified lineages of gastropods, the use of mitogenomes resulted in highly resolved phylogenies (*Williams, Foster & Littlewood, 2014*; *Osca, Templado & Zardoya, 2014*). Besides the use of the mitogenome in sequence-based phylogenies, mitogenome rearrangements can also provide phylogenetic signals

(*Grande, Templado & Zardoya, 2008*; *Stöger & Schrödl, 2013*; *Xie et al., 2019b*). Although the mitogenome is widely used in phylogeny of many gastropod groups (*Arquez, Colgan & Castro, 2014*; *Osca, Templado & Zardoya, 2014*; *Sevigny et al., 2015*; *Uribe, Zardoya & Puillandre, 2018*), there are limited numbers of reported stylommatophoran mitogenomes and phylogenetic studies in Stylommatophora in terms of usage of mitogenome sequence and rearrangement (*Romero, Weigand & Pfenninger, 2016*; *Xie et al., 2019a*; *Yang et al., 2019*). To date, complete or nearly complete mitogenomes have been reported for only 34 stylommatophoran species (NCBI, September, 2019).

In this study, we sequenced and annotated the complete mitogenome of *Arion vulgaris* Moquin-Tandon, 1855 (Gastropoda: Stylommatophora), which is considered as a serious invasive pest both in agriculture and private gardens. We compared it with the mitogenome of its congener *Arion rufus* (Linnaeus, 1758), and with all other previously reported stylommatophoran mitogenomes. We also reconstructed a phylogeny from stylommatophoran mitogenomes to estimate the phylogenetic position of *A. vulgaris* and to test the informativeness of mitogenome data in the reconstruction of Stylommatophora phylogeny. In addition, we obtained a dated phylogeny using this mitogenome dataset and fossil calibrations to estimate divergence times within Stylommatophora. Furthermore, selection analyses were performed to investigate the role of different selective forces shaping stylommatophoran mitogenomes. Finally, we compared the mitogenome organisations of stylommatophoran species using a comparative and phylogeny based method and tried to uncover the evolutionary pathways of mitogenome rearrangements.

## MATERIALS AND METHODS

### Specimen collection and DNA extraction

The specimen of *A. vulgaris* was collected from the garden of the Zoologische Staatssammlung München (ZSM), Germany. Total genomic DNA was extracted from mantle tissue using CTAB method (*Doyle & Doyle, 1987*).

### Mitogenome sequencing, annotation and analyses

The whole-genome sequencing was conducted with 150 bp pair-end reads on the Illumina Hiseq4000 Platform (Illumina, San Diego, CA) using 350 bp insert size libraries. Raw reads were processed by removing low quality reads, adapter sequences and possible contaminated reads using Fastp v0.20.0 (*Chen et al., 2018*) and Lighter v1.0.7 (*Song, Florea & Langmead, 2014*). In total, about 7.5G high quality base pairs of sequence data were obtained and the mitogenome was assembled using the MitoZ software (*Meng et al., 2019*), followed by manual curation using Geneious R9 (*Kearse et al., 2012*).

The annotation of tRNA genes of the *A. vulgaris* mitogenome was performed using MITOS (http://mitos.bioinf.uni-leipzig.de/index.py) (*Bernt et al., 2013b*) and ARWEN web servers (*Laslett & Canbäck, 2008*) based on their secondary structures and anticodon sequences. The locations and boundaries of PCGs and rRNA genes were identified manually by comparing with the *A. rufus* (KT626607) homologous gene sequences. The visualization of the secondary structure of tRNA genes was performed using VARNA v3-93 (*Darty, Denise & Ponty, 2009*) and RNAviz 2.0.3 (*De Rijk, Wuyts & De Wachter, 2003*). Intergenic

spacers and overlapping regions between genes were estimated manually. The largest non-coding region was defined as control region and the Mfold server (*Zuker, 2003*) was used to predict the secondary structure of this region. The "palindrome" tool within the European Molecular Biology Open Software Suite (EMBOSS) (*Rice, Longden & Bleasby, 2000*) was used for searching the palindromic sequences in the control region. Finally, the complete mitogenome of *A. vulgaris* was deposited in GenBank under accession number MN607980. The mitogenome of *A. vulgaris* is visualized using OrganellarGenomeDRAW (OGDRAW) (*Greiner, Lehwark & Bock, 2019*).

The nucleotide compositions, average nucleotide and amino acid sequence divergences and the relative synonymous codon usages (RSCU) of PCGs were computed using MEGA v7.0 (*Kumar, Stecher & Tamura, 2016*). The strand asymmetries were calculated according to the following formulas: AT-skew = $[A - T]/[A + T]$ and GC-skew = $[G - C]/[G + C]$ (*Perna & Kocher, 1995*).

## Phylogenetic and comparative analyses
### Alignment and model selection
Phylogenetic and comparative analyses were performed using the mitogenome dataset of 35 stylommatophoran species representing 18 families, and using one species from Systellommatophora, one species from Hygrophila, and one species from Ellobioidea as outgroups (Table 1). Each tRNA and rRNA gene was aligned individually using MAFFT (*Katoh & Standley, 2013*) algorithm in Geneious R9 (*Kearse et al., 2012*). The alignment of nucleotide sequences of each PCG was performed using MAFFT algorithm and the "translation align" option implemented in Geneious R9. The final alignment files were then concatenated using SequenceMatrix v.1.7.8 (*Vaidya, Lohman & Meier, 2011*). The optimal partitioning scheme and substitution models were inferred by PartitionFinder v1.1.1 (*Lanfear et al., 2012*) using the Bayesian Information Criterion (BIC) and the "greedy" algorithm with the option of "unlinked" branch lengths. The best-fit partitioning scheme and nucleotide substitution models were used in phylogenetic analyses (Table S1).

### Assessing the substitution saturation level
The substitution saturation levels in different genes and codon positions were estimated comparing the uncorrected p-distances and the distances calculated by applying the GTR + G + I evolutionary model selected based on the BIC using jModelTest v2.1.7 (*Darriba et al., 2012*). All genetic distances were computed with PAUP v4.0 b10 (*Swofford, 2002*).

### Phylogenetic reconstruction
Two different datasets were created for phylogenetic analyses to test the influence of saturated genes and codon positions: (1) 13 PCGs including all codon positions plus the 22 tRNAs and two rRNAs (P123RNA) and (2) PCGs excluding the five saturated genes and third codon positions, plus 22 tRNAs and two rRNAs (8P12RNA, Table S2). Maximum likelihood (ML) trees were constructed with RAxML v8.0.9 (*Stamatakis, 2014*) implemented in Geneious R9 applying the best-fit evolutionary model for each partition under 1,000 bootstrap replicates. For Bayesian Inference (BI) analyses, MrBayes v3.2.2 (*Ronquist et al., 2012*) was employed with two independent runs of 10 million generations

**Table 1    List of stylommatophoran mitogenomes used in phylogenetic and comparative analyses.**

|  | Species | Family | Accession number | References |
|---|---|---|---|---|
| STYLOMMATOPHORA | *Arion vulgaris* | Arionidae | MN607980 | This study |
|  | *Arion rufus* | Arionidae | KT626607 | *Romero, Weigand & Pfenninger (2016)* |
|  | *Achatinella fulgens* | Achatinellidae | MG925058 | *Price et al. (2018)* |
|  | *Achatinella mustelina* | Achatinellidae | NC030190 | *Price et al. (2016a)* |
|  | *Achatinella sowerbyana* | Achatinellidae | KX356680 | *Price et al. (2016b)* |
|  | *Partulina redfieldi* | Achatinellidae | MG925057 | *Price et al. (2018)* |
|  | *Achatina fulica* | Achatinidae | KM114610 | *He et al. (2016)* |
|  | *Deroceras reticulatum* | Agriolimacidae | NC035495 | *Ahn et al. (2017)* |
|  | *Aegista aubryana* | Bradybaenidae | NC029419 | *Yang et al. (2016)* |
|  | *Aegista diversifamilia* | Bradybaenidae | NC027584 | *Huang, Lin & Wu (2015)* |
|  | *Dolicheulota formosensis* | Bradybaenidae | NC027493 | *Huang, Lin & Wu (2015)* |
|  | *Mastigeulota kiangsinensis* | Bradybaenidae | NC024935 | *Deng et al. (2016)* |
|  | *Camaena cicatricosa* | Camaenidae | NC025511 | *Wang et al. (2014)* |
|  | *Camaena poyuensis* | Camaenidae | KT001074 | Unpublished |
|  | *Cerion incanum* | Cerionidae | NC025645 | *González et al. (2016)* |
|  | *Cerion tridentatum costellata* | Cerionidae | KY249249 | Unpublished |
|  | *Cerion uva* | Cerionidae | KY124261 | *Harasewych et al. (2017)* |
|  | *Albinaria caerulea* | Clausiliidae | NC001761 | *Hatzoglou, Rodakis & Lecanidou (1995)* |
|  | *Gastrocopta cristata* | Gastrocoptidae | NC026043 | Unpublished |
|  | *Cernuella virgata* | Geomitridae | NC030723 | *Lin et al. (2016)* |
|  | *Helicella itala* | Geomitridae | KT696546 | *Romero, Weigand & Pfenninger (2016)* |
|  | *Cepaea nemoralis* | Helicidae | NC001816 | *Yamazaki et al. (1997)* |
|  | *Cylindrus obtusus* | Helicidae | NC017872 | *Groenenberg et al. (2012)* |
|  | *Cornu aspersum* | Helicidae | NC021747 | *Gaitán-Espitia, Nespolo & Opazo (2013)* |
|  | *Helix pomatia* | Helicidae | NC041247 | *Korábek, Petrusek & Rovatsos (2019)* |
|  | *Orcula dolium* | Orculidae | NC034782 | *Groenenberg et al. (2017)* |
|  | *Naesiotus nux* | Orthalicidae | NC028553 | *Hunter et al. (2016)* |
|  | *Meghimatium bilineatum* | Philomycidae | NC035429 | *Xie et al. (2019a)* and *Xie et al. (2019b)* |
|  | *Philomycus bilineatus* | Philomycidae | MG722906 | *Yang et al. (2019)* |
|  | *Polygyra cereolus* | Polygyridae | NC032036 | Unpublished |
|  | *Praticolella mexicana* | Polygyridae | KX240084 | *Minton et al. (2016)* |
|  | *Pupilla muscorum* | Pupillidae | NC026044 | Unpublished |
|  | *Succinea putris* | Succineidae | NC016190 | *White et al. (2011)* |
|  | *Microceramus pontificus* | Urocoptidae | NC036381 | Unpublished |
|  | *Vertigo pusilla* | Vertiginidae | NC026045 | Unpublished |
| Ellobioidea | *Carychium tridentatum* | Ellobiidae | KT696545 | *Romero, Weigand & Pfenninger (2016)* |
| Hygrophila | *Galba pervia* | Lymnaeidae | NC018536 | *Liu et al. (2012)* |
| Systellommatophora | *Platevindex mortoni* | Onchidiidae | GU475132 | *Sun et al. (2016)* |

with four Markov chains (three cold, one heated), sampling every 1,000 generations and a burn-in of 25% trees. The stationarity of the chains was assessed using the program Tracer v1.7 (*Rambaut et al., 2018*). The consensus phylogenetic trees were visualized using FigTree v1.4.0 (*Rambaut, 2012*).

### Divergence time estimation

MCMCTree program implemented in the Phylogenetic Analysis by Maximum Likelihood (PAML) package v4.9 (*Yang, 2007*) was used for Bayesian estimation of divergence times of each species. Substitution rate per site was estimated by BASEML and was used to set the prior for the mean substitution rate in the Bayesian analysis. MCMC was run by 50 × 10,000 iterations with the REV substitution model. The soft bounds of *Helix pomatia* + *Cornu aspersum* [divergence time between 34 million years ago (Ma) and 42 Ma], *Mastigeulota kiangsinensis* + (*Dolicheulota formosensis* + (*Aegista aubryana* + *Aegista diversifamilia*)) (divergence time between 25 Ma and 51 Ma), and *Camaena cicatricosa* + *Camaena poyuensis* (divergence time between 16 Ma and 39 Ma) were used as external calibrations (*Razkin et al., 2015*) and the estimated nodal age of Tectipleura [244 Ma (210–279 Ma)] was used for the calibration of the root (*Kano et al., 2016*).

### Selection analyses

The CODEML implemented in PAML was used to estimate the ratio of nonsynonymous/synonymous substitution rate ($\omega$ = dN/dS) and to explore the role of different selective constraints working on each PCG under the one-ratio model (Model A: model = 0, NSsites = 0, fix_omega = 0, omega = 1). Gaps and ambiguous sites of sequence alignments were included in the analyses. For each PCG, likelihood ratio tests (LRTs) were used to compare the null neutral model (Model B: model = 2, NSsites = 2, fix_omega = 1, omega = 1) against alternative models of branch-specific positive selection (Model C: model = 2, NSsites = 2, fix_omega = 0, omega = 1.5). The Bayes Empirical Bayes (BEB) algorithm in CODEML was used to detect the positively selected sites. Furthermore, the adaptive branch-site random effects likelihood (aBSREL) (*Smith et al., 2015*) implemented in DATAMONKEY webserver (*Weaver et al., 2018*) was used to search the signatures of episodic positive diversifying selection testing each branch. In addition, mixed effects model of evolution (MEME) (*Murrell et al., 2012*) was used to detect episodic or diversifying selection at individual sites and a fast, unconstrained Bayesian approximation for inferring selection (FUBAR) (*Murrell et al., 2013*) was used for providing additional support to the detection of sites evolving under positive or negative selection. Each PCG was also evaluated in terms of properties and magnitude of amino acid changes using TreeSAAP v3.2 (*Woolley et al., 2003*), which uses 31 properties of amino acids and categorizes the degree of substitutions to eight categories (1–8).

### Comparison of mitogenome organizations

Mitogenome organizations and gene rearrangements of stylommatophoran species were analysed via the CREx web server (http://pacosy.informatik.uni-leipzig.de/crex) (*Bernt et al., 2007*). The gene orders of ancestral nodes were reconstructed using the Maximum Likelihood for Gene Order Analysis (MLGO, http://geneorder.org/) (*Hu, Lin & Tang, 2014*) with the input tree obtained by phylogenetic approaches, and the orders of the protein coding, rRNA and tRNA genes were compared with the inferred ancestral mitogenomes. A distance matrix was calculated based on number of common intervals, and the output diagram visually examined to identify shared and/or derived gene rearrangements as well as mechanisms of rearrangements.

## RESULTS AND DISCUSSION

### Mitogenome characteristics and nucleotide composition

The complete mitogenome sequence of *A. vulgaris* was obtained with a length of 14,547 bp (Table 2) and its size was within the range of the those of other reported stylommatophoran mitogenomes, varying between 13,797 bp in *Camaena poyuensis* and 16,879 bp in *Partulina redfieldi* (*Price et al., 2018*). It includes the entire set of 37 mitochondrial genes: 13 PCGs, 22 tRNAs and two rRNAs. Twenty-four genes were located on the J strand, while the remainings were encoded by the opposite N strand (Table 2, Fig. 1).

The nucleotide composition of *A. vulgaris* mitogenome was distinctly biased towards A and T, with a 70.20% A + T content, and comparable to other reported stylommatophoran mitogenomes, varying between 59.79% A + T in *Cepea nemoralis* (*Yamazaki et al., 1997*) and 80.07% A + T in *Achatinella mustelina* (*Price et al., 2016a*) (Table 3 and Table S3). A bias towards A and T nucleotides was also observed in PCGs of the *A. vulgaris* mitogenome with a 69.34% A + T content (Table 3). The A + T content of the 3rd codon position (79.64%) was higher than those of the 2nd (64.21%) and 1st codon positions (64.18%). Similar to other reported stylommatophoran mitogenomes (Table S3), the AT- and GC-skews were found slightly negative (−0.0756) and positive (0.0431) in the whole mitogenome of *A. vulgaris*, respectively. A pronounced T and G skew was also observed in all PCGs (−0.1508, 0.0472), PCGs on the majority strand (−0.1447, 0.0596), and tRNA genes (−0.0010, 0.1582) (Table 3). The T- and G-skewed mitogenome of *A. vulgaris* might be explained by the spontaneous deamination of cytosine during replication and transcription processes (*Reyes et al., 1998*). The PCGs encoded on the minority strand displayed a T- and C-skewed pattern (−0.1783 AT-skew, −0.0065 GC-skew), contrary to the expected high rates of Ts and Gs on the minority strand for most of the metazoans (*Hassanin, Léger & Deutsch, 2005*).

### Protein coding genes and codon usage

In comparison, the lengths of the PCGs of *A. vulgaris* mitogenome were within the range of those of other stylommatophoran mitochondrial PCGs. The *ND4* gene was the most variable gene in length and has a variability of 53 codons among stylommatophorans (419 codons in *Microceramus pontificus* and 472 codons in *Orcula dolium*). The most conserved gene in length was *COX1* and it exhibits variability with only 16 codons between species of Stylommatophora (501 codons in *Achatinella mustelina* and 517 codons in *Achatina fulica*). Compared with the mitogenome of *A. rufus*, the lengths of PCGs of *A. vulgaris* were distinct except for *COX1*, *COX2*, *CYTB* and *ND1* genes. The *ND6* gene was the most variable gene in length and was longer in the *A. vulgaris* mitogenome by 11 codons. Based on the amino acid identities, the most conserved PCG was *COX1* (56.45%) whereas the least conserved was *ND6* (11.92%) among the stylommatophoran mitogenomes. The most conserved PCG was *COX1* (97.45%) whereas the least conserved was *ATP8* (68.18%) based on the amino acid identities between the two *Arion* mitogenomes (Table S4).

In the *A. vulgaris* mitogenome, most of the PCGs initiated with typical ATN start codon, except for *COX1*, *ND5* and *ATP8* genes which use TTG, ACA and GTG triplets as start codons, respectively (Table 2). The TTG and GTG start codons are also accepted

**Table 2  Mitogenome summary of *Arion vulgaris*.**

| Gene | Strand | From | To | Size | Start codon | Stop codon | Anticodon | IGN |
|---|---|---|---|---|---|---|---|---|
| COX1 | J | 1 | 1,530 | 1,530 | TTG | TAG | | 4 |
| tRNA-Val | J | 1,535 | 1,600 | 66 | | | UAC | 0 |
| 16S rRNA | J | 1,601 | 2,613 | 1,013 | | | | 0 |
| tRNA-Leu | J | 2,614 | 2,675 | 62 | | | UAG | 11 |
| tRNA-Pro | J | 2,687 | 2,752 | 66 | | | UGG | 13 |
| tRNA-Ala | J | 2,766 | 2,831 | 66 | | | UGC | 7 |
| ND6 | J | 2,839 | 3,312 | 474 | ATG | TAG | | −41 |
| ND5 | J | 3,272 | 4,960 | 1,689 | ACA | TAA | | −10 |
| ND1 | J | 4,951 | 5,853 | 903 | ATG | TAG | | 15 |
| ND4L | J | 5,869 | 6,163 | 295 | ATA | T- | | −15 |
| CYTB | J | 6,149 | 7,228 | 1,080 | ATG | TAA | | −2 |
| tRNA-Asp | J | 7,227 | 7,296 | 70 | | | GUC | 10 |
| tRNA-Cys | J | 7,307 | 7,363 | 57 | | | GCA | 0 |
| tRNA-Phe | J | 7,364 | 7,425 | 62 | | | GAA | 0 |
| COX2 | J | 7,426 | 8,094 | 669 | ATG | TAG | | 1 |
| tRNA-Trp | J | 8,096 | 8,160 | 65 | | | UCA | 91 |
| tRNA-Tyr | J | 8,252 | 8,315 | 67 | | | GUA | 0 |
| Control region | J | 8,316 | 8,685 | 370 | | | | 0 |
| tRNA-Gly | J | 8,686 | 8,763 | 78 | | | UCC | −20 |
| tRNA-His | J | 8,744 | 8,809 | 66 | | | GUG | −3 |
| tRNA-Glu | N | 8,807 | 8,873 | 67 | | | UUC | 5 |
| tRNA-Gln | N | 8,879 | 8,942 | 64 | | | UUG | 0 |
| 12S rRNA | N | 8,943 | 9,689 | 747 | | | | 0 |
| tRNA-Met | N | 9,690 | 9,754 | 65 | | | CAU | 0 |
| tRNA-Leu | N | 9,755 | 9,820 | 66 | | | UAA | −32 |
| ATP8 | N | 9,789 | 9,971 | 183 | GTG | TAA | | 0 |
| tRNA-Asn | N | 9,972 | 10,033 | 62 | | | GUU | −8 |
| ATP6 | N | 10,026 | 10,688 | 663 | ATA | TAA | | −9 |
| tRNA-Arg | N | 10,680 | 10,746 | 67 | | | UCG | 3 |
| ND3 | N | 10,750 | 11,094 | 345 | ATG | TAA | | 13 |
| tRNA-Ser2 | N | 11,108 | 11,176 | 69 | | | UGA | 49 |
| tRNA-Ser1 | J | 11,226 | 11,283 | 58 | | | GCU | 36 |
| ND4 | J | 11,320 | 12,633 | 1,314 | ATA | TAG | | −18 |
| tRNA-Thr | N | 12,616 | 12,681 | 66 | | | UGU | 0 |
| COX3 | N | 12,682 | 13,462 | 781 | ATG | T- | | 41 |
| tRNA-Ile | J | 13,504 | 13,567 | 64 | | | GAU | 1 |
| ND2 | J | 13,569 | 144,86 | 918 | ATG | TAA | | 0 |
| tRNA-Lys | J | 14,487 | 6 | 67 | | | UUU | −6 |

**Notes.**
J, major; N, minor; IGN, intergenic nucleotides.
Minus indicates overlapping sequences between adjacent genes.

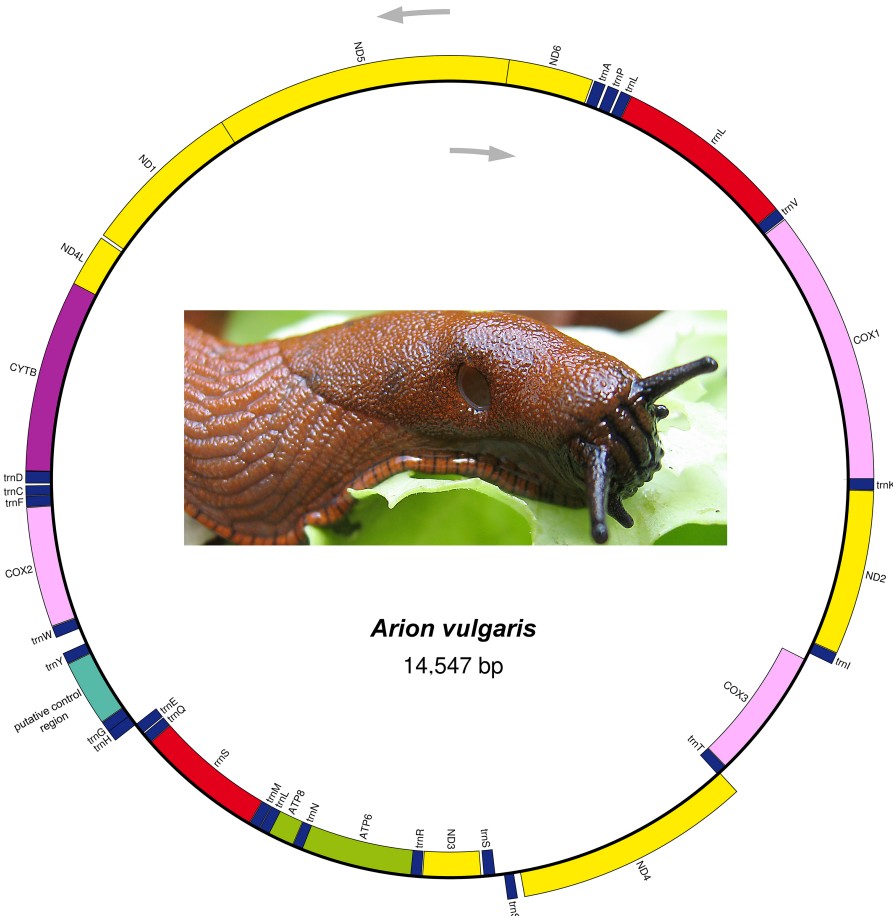

**Figure 1** **Mitogenome organization of *Arion vulgaris*.** Genes transcribed from the J and N strands are shown outside and inside of the circle, respectively. PCGs coding complex I, complex III, complex IV and complex V components are marked with yellow, purple, pink and green, respectively. rRNA genes are coloured with red and the putative control region is coloured with cyan, while tRNA genes are coloured with dark blue and labelled by the single letter amino acid code.

as canonical start codons for invertebrate mitogenomes (*Yang et al., 2019*), however, ACA as start codon for *ND5* gene was reported for the first time for stylommatophoran mitogenomes. Most of the PCGs were inferred to use TAR as termination codon, except for *ND4L* and *COX3* which have an abbreviated T-termination codon and their products are probably completed via post-transcriptional polyadenylation (*Anderson et al., 1981*; *Ojala, Montoya & Attardi, 1981*).

The most frequently used amino acids by the PCGs of the mitogenomes of *A. vulgaris* and *A. rufus* were leucine (16.71% and 15.91% respectively) and serine (10.33% and 10.18% respectively), similar to PCGs of the mitogenome of other stylommatophoran species (Leu 16,60%, Ser 10.21% on average). The codons rich in A and T, such as UUA-Leu, AUU-Ile, UUU-Phe, AUA-Met, UAU-Tyr, were the most frequently used codons in all stylommatophoran mitochondrial PCGs. The codons rich in terms of G and C content,

**Table 3  Nucleotide composition of the *Arion vulgaris* mitogenome.**

| Feature | T% | C% | A% | G% | A + T% | AT-skew | GC-skew |
|---|---|---|---|---|---|---|---|
| Whole mitogenome | 37.75 | 14.26 | 32.45 | 15.54 | 70.20 | −0.076 | 0.043 |
| Protein coding genes | 39.90 | 14.61 | 29.44 | 16.05 | 69.34 | −0.151 | 0.047 |
| First codon position | 33.54 | 14.01 | 30.64 | 21.81 | 64.18 | −0.045 | 0.218 |
| Second codon position | 45.65 | 19.42 | 18.55 | 16.38 | 64.21 | −0.422 | −0.085 |
| Third codon position | 40.50 | 10.39 | 39.14 | 9.97 | 79.64 | −0.017 | −0.021 |
| Protein coding genes-J | 39.78 | 14.34 | 29.72 | 16.16 | 69.50 | −0.145 | 0.060 |
| First codon position-J | 32.94 | 13.93 | 31.23 | 21.90 | 64.17 | −0.027 | 0.222 |
| Second codon position-J | 45.91 | 19.07 | 18.53 | 16.49 | 64.44 | −0.425 | −0.073 |
| Third codon position-J | 40.50 | 10.01 | 39.41 | 10.08 | 79.90 | −0.014 | 0.003 |
| Protein coding genes-N | 40.42 | 15.80 | 28.19 | 15.60 | 68.60 | −0.178 | −0.006 |
| First codon position-N | 36.24 | 14.37 | 27.98 | 21.41 | 64.22 | −0.129 | 0.197 |
| Second codon position-N | 44.50 | 20.95 | 18.65 | 15.90 | 63.15 | −0.409 | −0.137 |
| Third codon position-N | 40.52 | 12.08 | 37.92 | 9.48 | 78.44 | −0.033 | −0.121 |
| tRNA genes | 36.40 | 11.48 | 36.33 | 15.80 | 72.72 | −0.001 | 0.158 |
| rRNA genes | 33.24 | 13.58 | 38.18 | 15.00 | 71.42 | 0.069 | 0.050 |
| Control region | 38.92 | 18.65 | 30.81 | 11.62 | 69.73 | −0.116 | −0.232 |

CGC-CGG-Arg, CAG-Gln, UGC-Cys, CUC-Leu and UCG-Ser were rarely used in both *Arion* mitogenomes (Table S5, Fig. 2). CGN-Arg, CCS-Pro, GCS, UCG and UGC codons are seldom used or never used also in the stylommatophoran mitogenomes and reflected a significant relationship between codon usage and nucleotide content (Table S5).

## tRNA and rRNA genes

All of the tRNA genes could be folded into a usual clover-leaf secondary structure, except for *trnS1* (AGN) and *trnC* which lacked dihydrouridine (DHU) and TΨC arms, respectively and formed simple loops (Fig. S1). Their lengths ranged between 57 bp (*trnC*) and 78 bp (*trnG*), with an average 72.72% A + T content. 26 mismatched positions were observed in stem regions and all of the mismatches were G–U pairs (Fig. S1).

The exact boundaries of rRNA genes were determined as being bounded by the adjacent tRNA genes. The *rrnL* gene was located between *trnV* and *trnL1* genes, and the *rrnS* gene was located between *trnQ* and *trnM* genes. The length of the *rrnL* gene was 1,013 bp, with a 71.17% A + T content, while that of *rrnS* gene was 747 bp, with a 71.75% A + T content. These were comparable in ranges to homologous genes in other reported stylommatophoran species, ranging from 605 to 1215 bp in *rrnL* and from 564 to 857 bp in *rrnS*.

## Non-coding and overlapping regions

The total length of intergenic regions in the *A. vulgaris* mitogenome was 670 bp in 16 locations ranging between 1 and 370 bp (Table 2). In general, the largest non-coding region in the animal mitogenomes is considered to contain the signals for replication and transcription, and so called as the control region (*Wolstenholme, 1992*). The possible candidate for the control region in *A. vulgaris* mitogenome was the largest non-coding

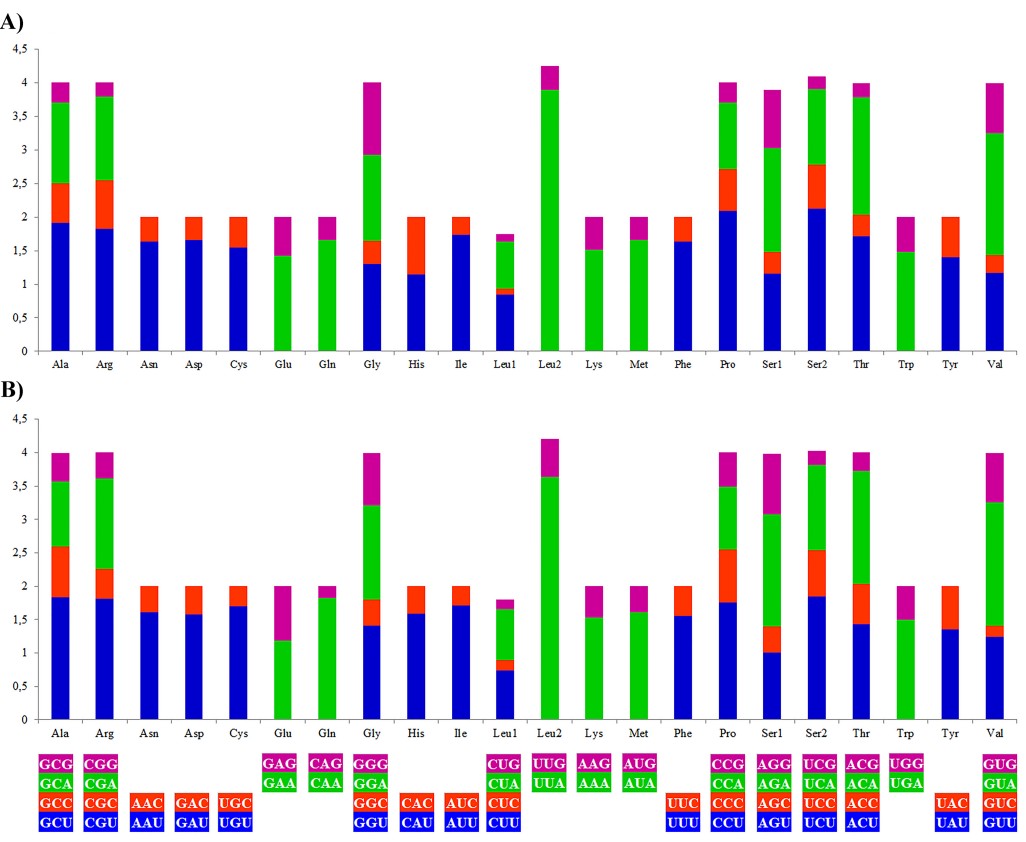

**Figure 2** **Relative synonymous codon usage (RSCU) of the (A)** *A. vulgaris* **and (B)** *A. rufus* **mitogenomes.** Codon families are provided on the *x* axis. The stop codons are not given.

region located between *trnY* and *trnG genes* with 370 bp in length. This sequence did not give BLAST hits with other putative CRs of other molluscan mitogenomes, however a part of the sequence with 67 bp in length displayed 79.11% sequence similarity with the mitochondrial control region of an amphibian species (*Indotyphlus maharashtraensis*, KF540157). Nucleotide composition of this region was slightly biased towards A + T with a 69.73% A + T content. The putative control region had a nine bp poly-T stretch and formed a stable secondary structure comprising seven stems and loops (Fig. 3). Furthermore, this sequence also contained a lot of palindromic sequences which are varying between 4 and 8 bp, but tandemly repeated sequences were not found.

The second largest non-coding region was found between *trnW* and *trnY* with a length of 91 bp (Table 2). The A + T composition of the sequence was higher than that of whole genome and putative control region with an 86.81% A + T. This non-coding region also contained a seven bp poly-A stretch and was folded into a secondary structure with two stem and loops. This secondary structure forming AT-rich sequence might function as the origin of the second strand (*Wolstenholme, 1992*).

Eleven overlapping regions with a total length of 164 bp were found throughout the mitogenome of *A. vulgaris*. The largest overlapping region was 41 bp in length and located

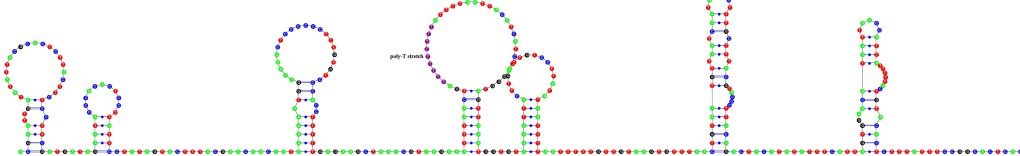

**Figure 3   Predicted secondary structure of putative control region of *A. vulgaris* mitogenome.** Nucleotides are coloured as follows: Adenine is green, thymine is red, cytosine is blue and guanine is black. The poly-T stretch is labelled with purple.

between *ND6* and *ND5* genes, while the second largest was 32 bp and located between *trnL2* and *ATP8* (Table 2).

## Phylogeny and divergence times of stylommatophoran species

Regression analyses of pairwise distances revealed that the 1st and 2nd codon positions of *ATP8*, *ND2*, *ND3*, *ND4L* and *ND6* genes, as well as the 3rd codon positions of all PCGs were saturated (Table S2). Four phylogenetic reconstruction analyses were performed with combination of inference methods and different data matrices to test the influence of inference methods and saturation level of genes/codon positions on tree topology and nodal support. Three different tree topologies were obtained as the results of these analyses, and topologies were sensitive to both inference methods and exclusion of saturated genes/codon positions (Fig. 4 and Figs. S2–S4). Nodal support values were always higher in BI trees than ML trees of the corresponding dataset. The usage of all mitochondrial genes and codon positions (P123RNA dataset) under both approaches resulted with identical tree topology (Figs. S3 and S4), which were similar to the results of *Yang et al. (2019)* obtained using only amino acid sequences of mitochondrial PCGs. The results of these analyses supported the monophyly of all included superfamilies with high nodal supports except for the superfamily Helicoidea which recovered with low nodal support [Bayesian Posterior Probability (BPP) = 0.75, Bootstrap support (BS) = 58%] and recovered Arionoidea superfamily as sister group to Urocoptoidea + (Polygyroidea + Helicoidea) clade (BPP = 1.00, BS = 100%), and Succineoidea + Orthalicoidea clade was recovered as sister group to Arionoidea + (Urocoptoidea + (Polygyroidea + Helicoidea)) (BPP = 1.00, BS = 50%). The ML and BI analyses performed using the dataset constructed with the removal of the saturated PCGs and codon positions (8P12RNA) resulted in two different tree topologies (Fig. 4 and Fig. S2). The phylogenetic tree obtained from ML analysis did not support the monophyly of the superfamily of Helicoidea and the superfamily of Polygyroidea placed within the superfamily of Helicoidea (BS = 61%, Fig. S2). A highly resolved tree with higher nodal support values was obtained from the BI approach of the dataset 8P12RNA, and hence considered as most reliable tree for discussion. The results confirmed the taxonomic position of *A. vulgaris* as sister species to *A. rufus* and recovered the monophyly of the Arionoidea superfamily (Arionidae + Philomycidae) with high support values (BPP = 1.00). A well-supported sister group relationship between Arionoidea and Orthalicoidea was recovered (BPP = 0.98) for the first time. However, previous studies using different datasets and sampling of taxa have proposed different sister groups with Arionoidea superfamily.

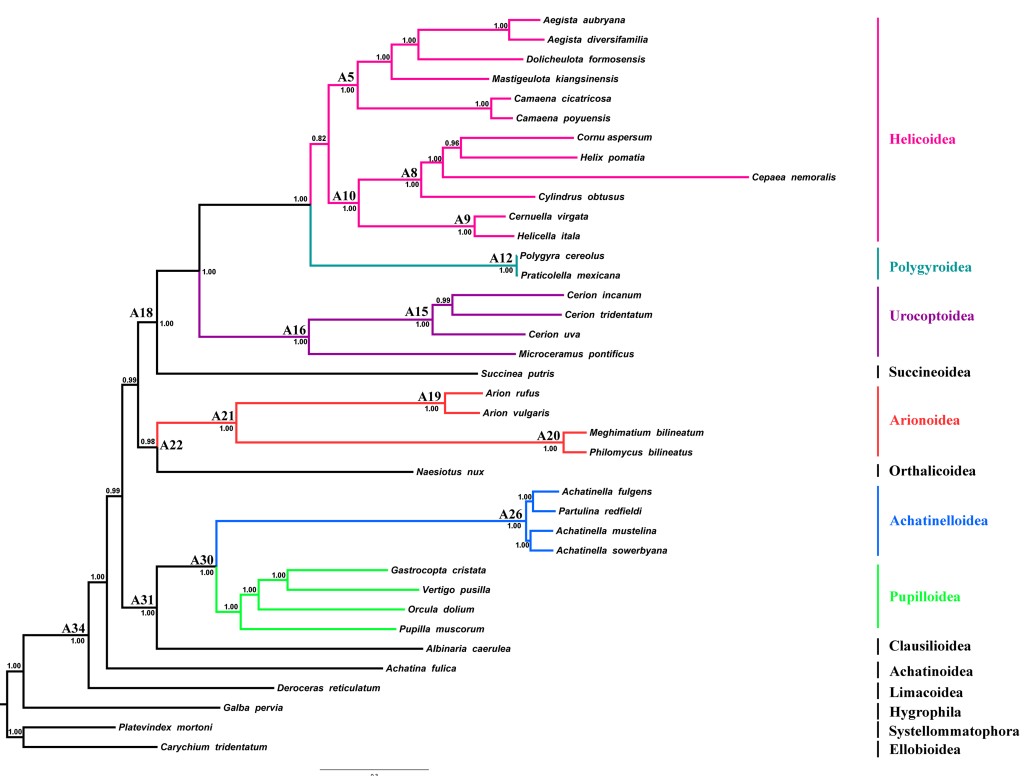

**Figure 4** Stylommatophoran phylogenetic tree constructed under BI using the dataset 8P12RNA. *Carychium tridentatum* (Ellobioidea), *Platevindex mortoni* (Systellommatophora) and *Galba pervia* (Hygrophila) were used as outgroup. Nodes are labelled with numbers refer to hypothetical ancestral mitogenome organizations inferred by MLGO.

*Wade, Mordan & Naggs (2006)* have found the superfamily Limacoidea as sister group to the superfamily Arionoidea using 160 stylommatophoran species, however they used only 823 nucleotides from rRNA gene-cluster. *Holznagel, Colgan & Lydeard (2010)* have proposed a sister group relationship between Arionoidea and Limacoidea + Zonitoidea based on the 28S rRNA sequences using seven species from Stylommatophora. The sister group relationships between Arionoidea and Urocoptoidea + Enoidea + Helicoidea (*Jörger et al., 2010*) or Limacoidea + (Succineoidea + Helicoidea) (*Dayrat et al., 2011*) have also been suggested by previous studies using relatively longer DNA sequences, however in both studies, only five stylommatophoran species were included for phylogenetic analyses. Furthermore, *Xie et al. (2019b)* have proposed sister group relationship between Arionoidea and Succineoidea using only amino acid dataset of mitochondrial PCGs, and stated it might be an artefact of poor taxon sampling.

In the phylogenetic tree obtained from 8P12RNA under BI approach, the monophyly of all included families and superfamilies were also supported with high support values except for the superfamily Helicoidea which supported with a low nodal support (BPP = 0.82) (Fig. 4). Arionoidea + Orthalicoidea clade was recovered as sister group to Succineoidea + (Urocoptoidea + (Polygyroidea + Helicoidea)). The tree (Fig. 4) also recovered *Deroceras*
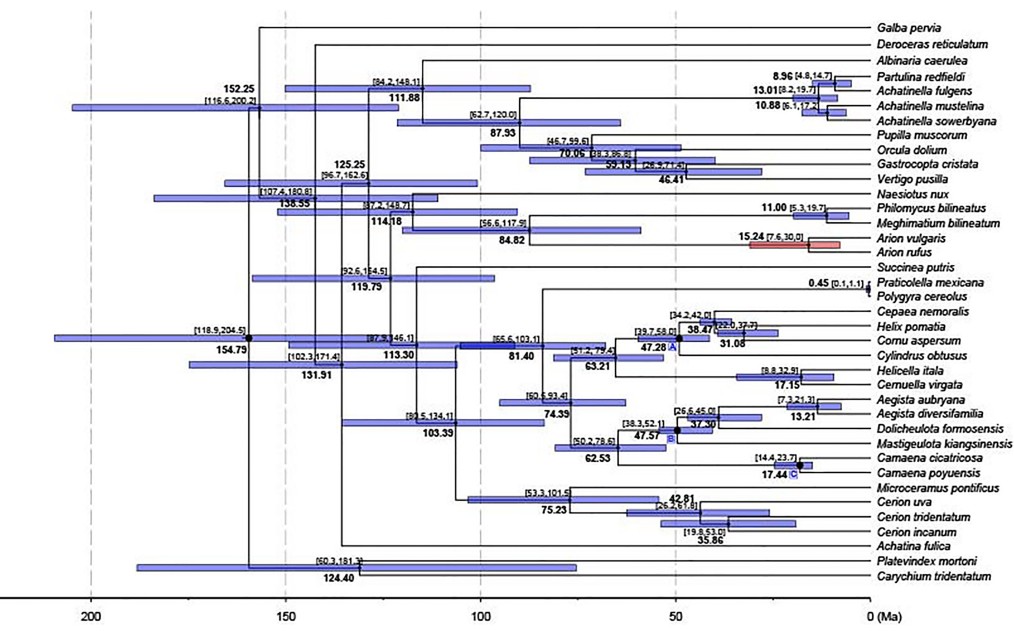

**Figure 5 Dated phylogenetic tree.** The axis on the bottom refers to million years. Letters in the boxes refers to external calibration points. The split between *A. vulgaris* and *A. rufus* was shown with a red bar, while the remainings were shown with blue bars.

*reticulatum* (Limacoidea: Agriolimacidae) at the most basal placement and did not support the monophyly of the suborder Helicina similar to the tree in *Yang et al. (2019)*.

A chronogram for Stylommatophora divergence times based on the obtained tree topology is shown in Fig. 5. According to our divergence time analysis, the crown age of stylommatophorans was estimated as 138.55 Ma (95% CI [180.8–107.4 Ma]) corresponding to Early Cretaceous. Our estimated times for initial diversification of Stylommatophora are slightly older but broadly congruent with the fossil records and previous studies (*Tillier, Masselmot & Tillirt, 1996*; *Jörger et al., 2010*; *Dinapoli & Klussmann-Kolb, 2010*). Although *Solem & Yochelson (1979)* suggested a Paleozoic origin for Stylommatophora, the widely accepted fossil records with recognizable taxa began from Late Cretaceous (*Bandel & Riedel, 1994*). The Cretaceous origin of stylommatophoran species was also suggested by sequence studies of 28S rDNA fragments by *Tillier, Masselmot & Tillirt (1996)*, of combined data of 18S, 28S, 16S rDNA and COI by *Dinapoli & Klussmann-Kolb (2010)* and *Jörger et al. (2010)*. The diversification of the stylommatophoran species may have been influenced by the explosive radiation of angiosperms and speciation by host-switching during Cretaceous (*Friis, Pedersen & Crane, 2010*).

The split time of *Achatina fulica* from other stylommatophoran species was inferred as 131.91 Ma in Early Cretaceous. The splits of the superfamilies Orthalicoidea and Arionoidea, of Succineoidea from Urocoptoidea + (Polygyroidea + Helicoidea), and of Clausilioidea + (Pupilloidea + Achatinelloidea) were dated to 114.18 Ma (95% CI [148.7–87.2 Ma]), 113.30 Ma (95% CI [146.1–87.9 Ma]) and 111.88 Ma (95% CI [148.1–84.2 Ma]), respectively, coinciding to the beginning of the Albian (Early Cretaceous). The

crown ages of the superfamilies Arionoidea, Urocoptoidea, Helicoidea and Pupilloidea were estimated corresponding to Late Cretaceous (84.82, 75.23, 74.39 and 70.06 Ma, respectively). The split of the two *Arion* species and the crown age of Achatinelloidea species were dated to 15.24 Ma (95% CI [30.0–7.6 Ma]) and 13.01 Ma (95% CI [19.7–8.2 Ma]), respectively, corresponding to the Miocene. The divergence time of *A. vulgaris* and *A. rufus* corresponds to one of Earth's most recent, global warming events, the Mid-Miocene Climatic Optimum (MMCO, 17–14.75 Ma) (*Böhme, 2003*). The MMCO is thought to have contributed to floristic and faunistic diversity across the world and so to animal-plant interactions, correlating with the rise in temperature (*Barnosky & Carrasco, 2002*; *Vicentini et al., 2008*; *Tolley, Chase & Forest, 2008*). The change of plant diversity, emergence of new host plants and the relative warm period may have triggered the diversification of the two *Arion* species. The divergence time of two polygyroid species was inferred as 0.45 Ma (95% CI [1.1–0.1 Ma]), in the Pleistocene.

## Selective pressures on stylommatophoran mitogenomes

The $\omega$ value for each of the 13 PCGs was inferred under one-ratio model using PAML and presented in Table 4. All of the $\omega$ values were extremely low ($\omega < 1$), ranging between 0.0129 for *COX1* and 0.2198 for *ATP8,* reflecting that all genes were under strong purifying selection consistent with the general mitogenome evolution pattern in animals (*Rand, 2001*; *Bazin, Glemin & Galtier, 2006*). Although purifying selection is the predominant selective force shaping stylommatophoran mitogenomes, the comparison of the null neutral model and alternative branch-specific positive selection model revealed six of the PCGs (*ATP6*, *COX2*, *COX3*, *ND2*, *ND4* and *ND5*) have variation in $\omega$ values along different branches. The variability in $\omega$ values indicated different selective forces acting on each gene as well as each branch. A more sensitive branch-site method, aBSREL, providing three states for each branch and allowing each site to evolve under any kind of the value ($<1$, 1 or $>1$) (*Smith et al., 2015*), was used for evaluating and confirming the selective forces across lineages determined by PAML analysis. All of the branches in the stylommatophoran phylogeny were tested with aBSREL analysis for each PCG, and the genes detected as under episodic diversifying selection were different from the results of branch-site model of PAML (Table 5) except for *COX3* and *ND4*. The aBSREL analyses discovered episodic diversifying selection in *ATP8* (at the branch leading to *Microceramus pontificus*), *COX1* (at the branch leading to *Achatinella mustelina*), *COX3* (at the branch leading to Arionoidea and the branch leading to *Philomycus bilineatus*), *ND3* (at the branch leading to *Helicella itala*), *ND4* (at the branch leading to *Succinea putris*) and *ND6* (at the branch leading to *Vertigo pusilla*). Due to their important function, mitochondrial genes might have a few positively selected sites and the signatures of purifying selection likely mask those of positive selection (*Meiklejohn, Montooth & Rand, 2007*; *Da Fonseca et al., 2008*). Therefore, two different methods were used to detect positive selection in addition to BEB analysis: FUBAR which estimates the rates of nonsynonymous and synonymous substitutions at each codon in a phylogeny, and MEME which estimates the probability for a codon to have experienced episodic positive selection and allows the $\omega$ ratio to vary across branches and codons. BEB analysis identified eight positively selected codons in total in three genes

**Table 4** Likelihood ratios of PAML analysis showing different selective pressures on the mitochondrial PCGs in Stylommatophora.

| Models[a] | | A | | B | | C | | A–B | | B–C | |
|---|---|---|---|---|---|---|---|---|---|---|---|
| Gene | $\omega$ | lnL[b] | Np[c] | lnL | Np | lnL | Np | LRT | P | LRT | P[d] |
| ATP6 | 0.0509 | −19658.6890 | | −19377.7446 | | −19374.7961 | | 561.8887 | 0.000 | −5.896982 | **0.015** |
| ATP8 | 0.2198 | −6460.1406 | | −6355.5745 | | −6355.5745 | | 209.1322 | 0.000 | −0.000006 | 1.000 |
| COX1 | 0.0129 | −26428.6827 | | −26169.9332 | | −26169.9332 | | 517.4990 | 0.000 | 0.000000 | 1.000 |
| COX2 | 0.0393 | −16022.1910 | | −15851.6973 | | −15848.6152 | | 340.9874 | 0.000 | −6.164250 | **0.013** |
| COX3 | 0.0317 | −18617.4781 | | −18345.8456 | | −18341.7284 | | 543.2650 | 0.000 | −8.234404 | **0.004** |
| CYTB | 0.0416 | −26893.7004 | | −26297.6496 | | −26297.6496 | | 1192.1017 | 0.000 | 0.000082 | 0.992 |
| ND1 | 0.0430 | −24857.6972 | 76 | −24528.7292 | 78 | −24529.0150 | 79 | 657.9359 | 0.000 | 0.571480 | 0.450 |
| ND2 | 0.0670 | −31769.5279 | | −31502.5684 | | −31498.5036 | | 533.9191 | 0.000 | −8.129668 | **0.004** |
| ND3 | 0.0607 | −11403.3678 | | −11183.4729 | | −11183.4729 | | 439.7899 | 0.000 | 0.000010 | 1.000 |
| ND4 | 0.0511 | −40498.0727 | | −40032.1913 | | −40026.0861 | | 931.7628 | 0.000 | −12.210500 | **0.000** |
| ND4L | 0.0727 | −10118.1495 | | −10076.4001 | | −10075.2708 | | 83.4989 | 0.000 | −2.258408 | 0.133 |
| ND5 | 0.0676 | −51970.2413 | | −51088.0681 | | −51092.3346 | | 1764.3464 | 0.000 | 8.533012 | **0.003** |
| ND6 | 0.1009 | −16916.1956 | | −16691.7006 | | −16691.7006 | | 448.9899 | 0.000 | 0.000018 | 1.000 |

**Notes.**
Degrees of freedom = 1.
[a]A, All branches have one $\omega$; B, All branches have same $\omega = 1$; C, Each branch has its own $\omega$.
[b]The natural algorithm of the likelihood value.
[c]Number of parameters.
[d]Bold faced figure indicate the statistical significance ($P < 0.05$).

(*ND2*, *ND4* and *ND5*), whereas FUBAR defined six positively selected codons in five genes (*ATP6*, *ATP8*, *COX2*, *CYTB* and *ND4L*). The MEME analysis found the signals of episodic positive selection at 22 codons in nine genes (*ATP6*, *ATP8*, *CYTB*, *ND2-6*, and *ND4L*). There was not any shared codon determined by all of the three analyses (Table 6). Only four codons in three genes were shared by the results of FUBAR and MEME analyses: 44th codon in *ATP8* gene, 12th codon in *CYTB* gene, and 13th and 57th codons in *ND4L* gene. Therefore, we focused only on these four codons in the TreeSAAP analyses. The positively selected substitution at codon 44 in *ATP8* gene was the change of TTA (Leu) to ATT (Ile) at branches leading to *M. kiangsinensis*, *Cerion incanum* and *Cerion uva*. This substitution was a radical chemical change with a magnitude category of 8 and had an impact on the increment of the equilibrium constant (ionization of COOH). The change at the codon 12 in *CYTB* gene was a conserved change with a magnitude category of 1 and was a substitution of TTG (Leu) to ATG (Met). The positively selected substitutions in *ND4L* gene were the change of ATT (Ile) to ATA (Met) at branch leading to *H. pomatia*, to GTT (Val) at branch leading to *C. nemoralis* at codon 13, and the change of TTT (Phe) to AAT (Asn) at branch leading to Arionidae family at codon 57. The substitution at the 13th codon was a radical change with a magnitude category of 8 altering the equilibrium constant (ionization of COOH), while that at the 57th codon was a radical change with a magnitude category of 7 and modifying the solvent accessibility of the protein.

Consequently, six positive selected genes (*ATP8*, *COX1*, *COX3*, *ND3*, *ND4* and *ND6*) detected by branch-specific aBSREL approach and three genes (*ATP8*, *CYTB* and *ND4L*) detected by codon-based BEB, FUBAR and MEME approaches were exposed to diversifying

**Table 5** Genes and branches detected to be exposed episodic diversifying selection using the aBSREL approach.

| Gene | Number of selected branches ($P < 0.05$) | Taxon | $\omega$ | Proportion of codons under selection |
|------|------|------|------|------|
| ATP8 | 1 | *Microceramus pontificus* | 288 | 0.460 |
| COX1 | 1 | *Achatinella mustelina* | 2,180 | 0.086 |
| COX3 | 2 | Arionoidea | 670 | 0.053 |
|      |   | *Philomycus bilineatus* | 119 | 0.092 |
| ND3 | 1 | *Helicella itala* | 49.4 | 0.220 |
| ND4 | 1 | *Succinea putris* | 4.18 | 0.370 |
| ND6 | 1 | *Vertigo pusilla* | 15.6 | 0.370 |

**Table 6** Genes/codons under diversifying or positive selection under codon-based models.

| Gene | BEB | FUBAR | MEME |
|------|------|------|------|
| ATP6 | – | 4 | 44 |
| ATP8 | – | 44 | 44, 57, 64, 92, 109 |
| COX1 | – | – | – |
| COX2 | – | 32 | – |
| COX3 | – | – | – |
| CYTB | – | 12 | 12 |
| ND1 | – | – | – |
| ND2 | 188 | – | 14, 16, 174 |
| ND3 | – | – | 27 |
| ND4 | 109, 170, 192, 301, 386, 427 | – | 9, 99 |
| ND4L | – | 13, 57 | 13, 57, 109, 111 |
| ND5 | 451 | – | 260, 501 |
| ND6 | – | – | 109, 179, 183 |

selection. Four of these genes (*ND3*, *ND4*, *ND4L* and *ND6*) play an important role in oxidative phosphorylation and are subunits of NADH dehydrogenase (Complex I) which is the most complicated and largest proton pump of the respiratory chain coupling electron transfer from NADH to ubiquinone. In addition to its important role in energy production, it has been shown that complex I is implicated in the regulation of reactive oxygen species (ROS) (*Sharma, Lu & Bai, 2009*). Substitutions in this complex might have been favoured for increasing the efficiency of proton pumping or regulating the response to ROS depending varying amount of oxygen in the atmosphere and adaptation to conditions in new habitats (temperature, humidity, altitude) and/or hosts. *CYTB* gene encodes only mitogenome derived subunit of Complex III and catalyses reversible electron transfer from ubiquinol to cytochrome c (*Da Fonseca et al., 2008*). The positively selected sites in complexes I and III have been suggested to contribute to environmental adaptation in different groups such as mammals, birds, fishes and insects (*Da Fonseca et al., 2008*; *Garvin, Bielawski & Gharrett, 2011*; *Garvin et al., 2014*; *Melo-Ferreira et al., 2014*; *Morales et al., 2015*; *Li et al., 2018*). In the cytochrome *c* oxidase complex (Complex IV), *COX1*

protein catalyses electron transfer to the molecular oxygen; *COX2* and *COX3* belong to the catalytic core of the complex may act as a regulator. *ATP8* gene encodes the part of ATP synthase (Complex V) regulating the assembly of complex (*Da Fonseca et al., 2008*). The favoured substitutions in *COX3* and *ATP8* gene might have an impact on assembly of the complexes IV and V. The positively selected substitutions and random accumulation of variation in mitochondrial PCGs of stylommatophoran species thus seem to be adaptive and affecting mitochondrial ATP production or protection from ROS effects, however effects of substitutions should be examined in a larger sample by considering protein folding and three-dimensional structure of complexes.

## Gene rearrangements in stylommatophoran mitogenomes

The ancestral mitogenome organisation of each node in the phylogeny was inferred using the maximum likelihood approach. The organisation of the hypothetical ancestral Stylommatophora mitogenome (node: A34, Fig. 4) was identical with that of *Deroceras reticulatum* as well as those of *Albinaria caerulea*, *Cernuella virgata* and *Helicella itala*. The mitogenome of *Achatina fulica* has only experienced the transposition of *trnP* to the downstream of *trnA* compared to its most recent ancestral mitogenome organisation. The common ancestors of Clausilioidea + (Pupilloidea + Achatinelloidea) (node: A31, Fig. 4), Orthalicoidea + Arionoidea (node: A22, Fig. 4), and Succineoidea + (Urocoptoidea + (Polygyroidea + Helicoidea)) (node: A18, Fig. 4) maintained the same order of hypothetical ancestral stylommatophoran mitogenome. In the mitogenome of the most recent common ancestor (MRCA) of Pupilloidea + Achatinelloidea (node: A30, Fig. 4), the reversal of *trnW*, *trnG* and *trnH* genes occurred individually and were followed by the reversal of the cluster *trnW-trnG-trnH*. In the superfamily Pupilloidea, rearrangements of several tRNA genes were observed: the transposition of the cluster *trnD-trnC* to downstream of *trnW* in *Pupilla muscorum*, transpositions of cluster *trnH-trnG* to downstream of *trnW* and of *trnT* to upstream of *COX3* in *Orcula dolium*, transposition of *trnG* to downstream of *trnW* in *Vertigo pusilla* and the reversal of *trnQ* in *Gastrocopta cristata*. In the mitogenome of the MRCA of the superfamily Achatinelloidea (node: A26, Fig. 4), *trnF-COX2-trnY-trnH-trnG-trnW-trnQ-ATP8-trnN-ATP6-trnR-trnE-rrnS-trnM* gene cluster rearranged as *trnW-trnQ-ATP8-ATP6-trnR-trnE-rrnS-trnM-trnF-COX2-trnY-trnH-trnG-trnN* via tandem duplication random loss (TDRL) mechanism. The organisation of the mitogenomes of achatinelloid species nearly matched with the putative ancestral order, except for *Achatinella sowerbyana* which has a transposed position of *trnK* to downstream of *ATP8* and a second copy of *trnL2*, and for *Partulina redfieldi* which has the inversion of *trnE* and *trnN* genes.

The mitogenome of *Naesiotus nux* has almost the same organisation with its MRCA (node: A22, Fig. 4), except for the second inverted copy of *ND4L* located between *trnL1* and *trnP*. The MRCA of the superfamily Arionoidea (node: A21, Fig. 4) had also identical mitogenome organisation with the ancestor of Stylommatophora, and the MRCAs of the families Arionidae (node: A19, Fig. 4) and of Philomycidae (node: A20, Fig. 4) were derived from this ancestor. The mitogenome of node A19 had shuffled positions of *trnY* and *trnW*, and also transpositions of *trnE* to downstream of *trnQ* and of *rrnS-trnM* to upstream

of *trnQ*. Both *Arion* mitogenomes also shared this mitogenome organisation and the rearranged orders of *trnW-trnY* and *trnE-trnQ-rrnS-trnM-trnL2-ATP8-trnN-ATP6-trnR* clusters seem to be synapomorphies of this genus. The mitogenome organisation of the node A20 was quite different from those of other stylommatophoran species, which had rearranged positions of almost all genes between *COX1* and *trnI* via two-steps TDRL, and two Philomycidae species also had identical organisation except for *Philomycus bilineatus* had a second copy of *trnC* located downstream of the original copy.

The mitogenome of *Succinea putris* has experienced the transpositions of *trnF* to upstream of *trnD* and of *trnW* to upstream of *trnY*, and also reverse transposition of the cluster *trnW-trnY* to the upstream of *ND3*. The MRCA of the all urocoptoid species (node: A16, Fig. 4) only had the reversal of *trnQ* gene from minor to major strand and *M. pontificus* has also maintained the identical arrangement. The four step requiring rearranged gene cluster was identified in the mitogenome of the MRCA of the genus *Cerion* (node: A15, Fig. 4): (i) reversal of *trnV-rrnL-trnL1*, (ii) reversal of *trnP*, (iii) reversal of *trnA*, and (iv) reversal of the cluster *trnL1-rrnL-trnV-trnP-trnA*. The mitogenome organisation remained the same in all three *Cerion* species and the rearranged state of *trnA-trnP-trnV-rrnL-trnL1* cluster might be a synapomorphy for this genus.

The MRCAs of the polygroid species (node: A12, Fig. 4) and Camaenidae + Bradybaenidae (node: A5, Fig. 4), as well as the *Polygyra cereolus* and *Praticolella mexicana,* had the transposed position of *trnG-trnH* to the upstream of *trnY*. The rearrangement of this cluster as *trnG-trnH-trnY* could be suggested as a synapomorphy for Polygyroidea, but more sampling is required to confirm its status at superfamily level. In the superfamily Helicoidea, the MRCAs of Geomitridae (node: A9, Fig. 4) and Geomitridae + Helicidae (node: A10, Fig. 4) shared the identical mitogenome organisation with the MRCA of Stylommatophora. Both of the Geomitridae species had also same mitogenome organisation except for *ATP8* in *Cernuella virgata,* in which this gene was missing, however it seems to be likely a misannotation. The mitogenome of MRCA of Helicidae species (node: A8, Fig. 4) had experienced the transpositions of *trnP* and cluster *trnT-COX3* to downstream of *ND6* and to upstream of *trnS1*, respectively. The mitogenome organisations of *Helix pomatia*, *Cornu aspersum* and *Cepaea nemoralis* have not changed and *trnA-ND6-trnP* and *trnS2-trnT-COX3-trnS1* gene orders might be interpreted as synapomorphic for these three species. However, the individual reversals of *trnA*, *ND6* and *trnP* genes followed by reversal of the cluster *trnA-ND6-trnP*, and reversal of *trnS1* were observed in *Cylindrus obtusus* mitogenome. In the mitogenomes of the species of the family Camaenidae, only the transpositions of *trnD* and *trnY* to downstream of *COX2* and to upstream of *trnG* were found, respectively. The arrangement of the *trnC-trnF-COX2-trnD-trnY-trnG* cluster could be considered as a synapomorphy for camaenid species, however the taxonomic level of this synapomorphy need to be evaluated in a wider taxonomic range. In the family of Bradybaenidae, the MRCA mitogenome had experienced only the reversal of *trnW*. In addition to this rearrangement, *Aegista* species also have the transposition of *ND3* gene to the downstream of *trnW* and the rearranged position of *ND3-trnW* cluster appears to be a synapomorphy for the genus.

## CONCLUSIONS

The sequencing and annotation of the mitogenome of *A. vulgaris* and its comparison with other stylommatophoran mitogenomes allow us to denote several conclusions: (i) the mitogenome characteristics of *A. vulgaris* are mostly consistent with the reported stylommatophoran mitogenomes; (ii) rearrangement events are detected in the *trnW-trnY* and *trnE-trnQ-rrnS-trnM-trnL2-ATP8-trnN-ATP6-trnR* gene clusters which may be apomorphic for the genus *Arion*, but further investigations are necessary; (iii) stylommatophoran mitogenome sequence information without the saturated positions seems to be useful for reconstructing phylogeny and estimating divergence times, and the taxon set used should be expanded; (iv) although purifying selection is the dominant force in shaping the stylommatophoran mitogenomes, in the background, several codons or different branches have experienced diversifying selection suggesting adaptation to new environmental conditions.

### Funding
This study has received funding from the European Union's Horizon 2020 research and innovation programme under the Marie Skłodowska-Curie grant agreement No 764840. The funders had no role in study design, data collection and analysis, decision to publish, or preparation of the manuscript.

### Grant Disclosures
The following grant information was disclosed by the authors:
European Union's Horizon 2020 research and innovation programme: 764840.

### Competing Interests
The authors declare there are no competing interests.

### Author Contributions
- Özgül Doğan and Zeyuan Chen conceived and designed the experiments, performed the experiments, analyzed the data, prepared figures and/or tables, authored or reviewed drafts of the paper, and approved the final draft.
- Michael Schrödl conceived and designed the experiments, authored or reviewed drafts of the paper, and approved the final draft.

### DNA Deposition
The following information was supplied regarding the deposition of DNA sequences:
The raw data is available at GenBank: MN607980.

### Data Availability
Raw data is available as Supplemental Files.

## Supplemental Information

Supplemental information for this article can be found online at http://dx.doi.org/10.7717/peerj.8603#supplemental-information.

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
