# Peer review of "The complete mitogenome of Arion vulgaris Moquin-Tandon, 1855 (Gastropoda: Stylommatophora): mitochondrial genome architecture, evolution and phylogenetic considerations within Stylommatophora"

_PeerJ, doi:10.7717/peerj.8603_

## Round 0.1 · original submission · Major Revisions

Two specialists in the field evaluated the present manuscript, and both have several concerns related to this paper. The reviewers have described all weak points found in this submission. Considering the evaluation carried out by both reviewers, I recommend a significant revision of this manuscript.

Reviewer 1 ·

Basic reporting

The authors indicate that accession number from GenBank is pending. If everything is correct, you get the number in less than a week (usually 48h). You shouldn’t submit the manuscript without the number because is one of the first quality reports that check if there are anything wrong with the data.

Experimental design

No comment

Validity of the findings

In general, I find the legends poor. Legends must be understandable by themselves. For some of them is necessary look into the main text or make interpretations (e.g. Fig. 5; Tables 1, 5 and 6).
On the other hand, I did not find the legends for the supplementary Figures S2, S3 and S4. Without legends I can’t say many things but, many clades were not supported… (Fig S2 & S4: Bootstrap support values? Lower than 70; Fig. S3: BPP ? lower than 0.95). Also in the Fig 4., at least the family Helicoidea is not supported (0.82 BPP) but in the text is indicated that was recovered with high support.

Additional comments

Please find all my comments in the pdf file attached.

Annotated reviews are not available for download in order to protect the identity of reviewers who chose to remain anonymous.

Reviewer 2 ·

Basic reporting

The manuscript meet the standards for publication.

Experimental design

The research question is well described but should be limited to the evolution of the stylommatophoran group. All the data analyses were well performed and the methods appropriate to their goals.

Validity of the findings

The authors have robust conclusions about the evolution of the taxonomic group and contribute to the understanding of natural selection over the mitogenome.

Additional comments

The manuscript describes evolution of the stylommatophoran group and the mitogenome architecture and selection. The authors sometimes describe and discuss the Arion vulgaris mitogenome, and sometimes the evolution of the entire taxonomic group. This is confusing and the fact they have sequenced the A. vulgaris mitogenome is a plus but not the main strength of the manuscript. I suggest limiting the entire manuscript to the phylogenetic reconstruction, selection and architecture of stylommatophorans. Therefore, I suggest change the title, the introduction is very informative about the diversification and evolution of the taxonomic group, but it should eliminate details about A. vulgaris, and add more the context of the natural selection over the mitogenome since it is only addressed in the objectives.

Lines 251-331: This three first subtitles of the Results and Discussion are very descriptive. Reduce significantly these sections. Most of this data are shown in tables at supplementary material but the caption could be more detail with a few more explanation and pointing at the tables the most important values. Sometimes the authors compare the Arion vulgaris mitogenome with all other species, and sometimes it does the comparison among all species (A. vulgaris as one of the species). Please limit to the second comparison.

The authors found three different tree topologies, but the authors describe the tree with higher support as the most reliable. The discussion between the tree found and the other studies are very important, but it is confusing here (lines 344-351). The authors should also discuss why those differences between their findings and the other studies (few or different markers, different phylogenetic reconstruction methods…).

The results are very relevant because they highlight the importance of eliminating saturated substitutions to perform phylogenetic reconstruction analyzes. In this regard, I think it is important to emphasize results such as:
1) phylogenies constructed with ML (Fig. S3) and BI (Fig S4) without eliminating substitutions and saturated genes share the same topology and place members of two different families as sister species (Succineoidea and Orthalicoidea)
2) When the phylogenies are constructed without the substitutions and saturated genes, the position of the species of the Succineoidea and Orthalicoidea family is recovered. ML places Polygyroidea within the Helicoid clade, but with a low node support (S2), while BI (Fig. 4) recovers the position of the Polygyroidea family.

402-404: This phrase is a bit confusing. It can be written more clearly.

438-439: The MEME and aBSREL methods evaluate the existence of episodic diversifying selection, that is, the selection arises sporadically within one or some lineages. However, FUBAR evaluates the existence of pervasive selection. In this sense, it is not correct to say that everyone suggests episodic diversifying selection. You could use only "diversifying selection" (Chek this in the abstract, lines 44-45).


Other minor reviews:
30-33: These two phrases are not necessary
Line 152: How did you obtain the mitogenome reads from the whole-genome reads?
Line 347: I am not clear what do you mean in this phrase, after “However”.
Figure 1 could be at supplementary. It is not the goal of the manuscript.
Line 386: Complex to write about the “diversification of Arion species” since it has been evaluated only two species.
Line 406: Eliminate “.”

---

## Round 0.2 · accepted · Accept

The authors revised their manuscript according to the comments made by the reviewers. In my view, the paper improved a lot and can be accepted as it is.

Reviewer 1 ·

Basic reporting

I think the authors made an excellent work with the revision of the manuscript. The authors answer and correct all the comments proposed by the reviewers in a clearly way. However, I have a minor consideration that should be corrected before accepting the ms. In one of the responses to the Reviewer 1, the authors say that they modified the text to indicate that the Helicoidea superfamily is not statistically supported but, I still can read in the abstract that the monophyly of all superfamilies is well supported. This should be removed from the abstract.
I hope my comments are clear and somehow helpful.

Experimental design

The goals of the work are well defined and all the analyses were correctly performed.

Validity of the findings

The conclusions of this manuscript are strong according to the results and will help in understand the evolutionary history of the stylommatophoran.

Additional comments

No comments